# Teleworking and Musculoskeletal Disorders: A Systematic Review

**DOI:** 10.3390/ijerph20064973

**Published:** 2023-03-11

**Authors:** Marc Fadel, Julie Bodin, Florence Cros, Alexis Descatha, Yves Roquelaure

**Affiliations:** 1Univ Angers, CHU Angers, Univ Rennes, Inserm, EHESP, Irset (Institut de Recherche en Santé, Environnement et Travail)—UMR_S 1085, IRSET-ESTER, SFR ICAT, F-49000 Angers, France; 2GRePS UR 4163, Université Lumière Lyon 2, F-69007 Lyon, France; 3Department of Occupational Medicine, Epidemiology and Prevention, Donald and Barbara Zucker School of Medicine, Hosftra University Northwell Health, New York, NY 11021, USA

**Keywords:** pain, musculoskeletal diseases, teleworking, public health, occupational health

## Abstract

Teleworking has spread drastically during the COVID-19 pandemic, but its effect on musculo-skeletal disorders (MSD) remains unclear. We aimed to make a qualitative systematic review on the effect of teleworking on MSD. Following the PRISMA guidelines, several databases were searched using strings based on MSD and teleworking keywords. A two-step selection process was used to select relevant studies and a risk of bias assessment was made. Relevant variables were extracted from the articles included, with a focus on study design, population, definition of MSD, confounding factors, and main results. Of 205 studies identified, 25 were included in the final selection. Most studies used validated questionnaires to assess MSD, six considered confounders extensively, and seven had a control group. The most reported MSD were lower back and neck pain. Some studies found increased prevalence or pain intensity, while others did not. Risk of bias was high, with only 5 studies with low/probably low risk of bias. Conflicting results on the effect of teleworking on MSD were found, though an increase in MSD related to organizational and ergonomic factors seems to emerge. Future studies should focus on longitudinal approaches and consider ergonomic and work organization factors as well as socio-economic status.

## 1. Introduction

The World Health Organization (WHO) and the International Labor Organization (ILO) define telework as: “the use of information and communications technology for work that is performed outside the employer’s premises” [1,2]. There are several terms related to this type of work, including remote work, which is the broadest term where the workplace can be anywhere outside the usual place, telework, which implies the usage electronic devises for remote work and home-based work/work from home (WFH), which can imply that the default working place is at home [1]. Hybrid work is a growing form of work which combines WFH and work in the office. In 2019, before the COVID-19 pandemic, 14% of workers in the European Union teleworked from home regularly or occasionally [3]. This number increased up to 40% of workers during summer 2020 following the multiple lockdowns and stabilized at 31% during spring 2022 [4]. The effect of teleworking on health seems to be contrasted both positively and negatively depending on the situations, with a predominant role of contextual factors [5,6,7]. Even before the pandemic happened, teleworking was implemented as a useful tool for allowing sustainable work and return to work, especially for workers with disabilities or who were suffering from chronic diseases, including cancer [8]. Indeed, traditional work environments may present barriers for these workers, especially those with cognitive limitations, and an adapted and familiar work environment at home may facilitate employment. Such accommodations require many changes in the work culture and vary significantly between countries [8]. Bouziri et al. highlighted the potential health impact during the COVID-19 pandemic, as well as possible recommendations for decreasing the new related risk [9]. For example, during containment, telework decreased the risk associated with transportation but also showed the lack of ergonomic measures for home workstations and the work environment in general. Indeed, office work can be associated with musculoskeletal disorders (MSD) caused by multiple interlinking factors defined by the concept of professional exposome, including the under-stimulation of the musculoskeletal system, comorbidities, and work-related organizational factors [10]. MSD is a broad term that is referenced at any affection of the soft periarticular tissues, following the EU-OSHA (European Agency for Safety and Health at Work) [11]. Medically codified diseases are included in this definition, like carpal tunnel syndrome or rotator cuff disease, but unspecific conditions like low back pain or neck pain are also included. Although there were some studies on the impact of teleworking on mental health before the COVID-19 pandemic [12], few of them focused on MSD [13,14].

The aim of this systematic review was to make a synthesis of the literature on the effect of teleworking on MSD. A qualitative approach was chosen, since heterogenous methods and results were expected.

## 2. Methods

### 2.1. Selection of Studies

A systematic review was conducted following the PRISMA (Preferred Reporting Items for Systematic Reviews and Meta-Analyses) guidelines [15]. The following databases were searched for studies relevant to the subject with the help of a librarian: PubMed, Web of Science, Embase, Cairn, EBSCO databases, and Google Scholar. Musculoskeletal disorders (and related) were searched with teleworking/working at home (and related, see Appendix A). Keywords related to MSD were chosen to engulf the various aspects of the EU-OSHA’s definition of MSD [11]. Only papers published since 1987 in English or French were included. Inclusion criteria were studies about workers (students were also included) exposed to teleworking/working at home, having an outcome related to MSD (whether pain, discomfort…), and having a quantitative approach. Studies about telemedicine and domestic work were excluded from the review, as well as non-published studies (conference abstract) and reviews. There were no other restrictions for the studies, particularly on study design (before/after study, control group, or no control group). After the exclusion of duplicates, studies were included in two steps: selection on title and abstract and selection on the full paper. At each step, all studies were included or excluded by two different reviewers among MF, JB, AD, and YR. In case of disagreement, a consensus was obtained by a third reviewer that was different from the first two. A cross-reference approach was also implemented to identify potentially relevant studies that may have been missed in the initial selection step. The extraction of studies was completed on 5 May 2022.

### 2.2. Extracted Data

The following variables were extracted from the articles included at the final selection step: first author, year of publication, country, objective of the study, design, population, recruitment and data collection method, exclusion criteria, number of subjects included, data of inclusion and potential follow-up, definition of MSD outcome, confounding factors considered, main results, and conclusion of the authors and limitations.

The risk of bias was assessed following the Navigation Guide for systematic reviews in environmental and occupational health [16]. For each study, two different reviewers among MF, JB, AD, and YR assessed the nine risk of bias domains, although two of them were not applicable for the observational studies (randomization and selecting outcome reporting). The last risk of bias domain (other risk of bias) that was adapted for this systematic review thus focused on two aspects: the representativeness of the studied population to the general teleworker population and the presence of a control group. In the case of disagreement, a consensus was obtained among all four reviewers.

## 3. Results

Of the 205 studies included, 87 were duplicates and 76 were deemed irrelevant according to the title and abstract. Seventeen more studies were excluded for various reasons detailed in the flow chart (Figure 1). In the end, 25 studies were selected. Table 1 summarizes the characteristics of the studies included. An exhaustive summary of the studies is available in Appendix A. All studies were published during and after 2020, and only one study used data from before the COVID-19 pandemic [12]. There was a relatively good representation of countries throughout the world, with most of them coming from Asia [17,18,19,20,21,22,23,24,25,26,27] and Europe [12,28,29,30,31,32,33,34], and some from America and Australia [35,36,37,38,39,40]. Only three studies did not have a cross-sectional design: the first was a before/after study [28], the second was a prospective cohort [37], and the last was an interventional study [19].

The majority of the populations included in the studies were office workers who had mandatory teleworking caused by lockdowns [17,18,19,23,24,25,27,28,29,31,32,33,34,35,36,37,38,40]. One study focused on participants already suffering from low back pain before implementation of telework [30], whereas others studies excluded participants with a history of MSK disorders or serious medical conditions [19,23,25,28,29,37]. The number of participants varied, ranging from 40 [19] to 12,774 [20]. Validated questionnaires were used in almost half of the studies included [17,18,19,28,29,31,34,35,37,38], with the most used one being the Nordic musculoskeletal questionnaire. Other outcomes included numerical scales and Likert type scales assessing the intensity and/or the frequency of pain or the evolution of pain before and after implementation of lockdowns (associated with WFH). Most articles inquired about pain from various region of the body, but some focused on either neck pain [12,28,37] or lower back pain [20,21,22,30].

Adjusted analyses assessing the effect of teleworking and MSD were conducted in eight studies [12,18,20,21,22,26,28,37]. Adjustment variables selected were often different, and related to demographic data (age, sex, education, income, marital status), lifestyle habits (smoking, alcohol, physical activity), comorbidities, general work factors (working hours management responsibility, occupational stress), and work factors specifically related to teleworking (working hours at home, frequency of telecommuting, ergonomic factors of the home office). Seven studies had a control group [12,20,21,26,28,32,40], which was a before intervention control for one study [28], or participants who did not WFH, or who had very few days teleworking.

The most reported MSD were low back pain and neck pain. Some studies did not find any association between MSD disorders and telework [12,24,28,29,30,34]. Others found an increase in prevalence or intensity of MSD pain during lockdowns (WFH) compared to before [17,18,23,25,33,36,38,39,40], or when comparing groups of different telework frequency [20,22,25,26,32,40]. Certain ergonomic factors and psychosocial factors were associated with increased intensity or frequency of musculoskeletal pain. [12,18,23,24,28,32,33,35,36,38]. For example, having an appropriate location to WFH with sufficient space and less demanding workloads decreased the effect of WFH on MSD.

Table 2 shows the risk of bias assessment for all the studies included. Overall, the risk of bias for the studies was high. Only five studies had low or probably low risk of bias for the specific criteria in this review, which were the representativeness of the population and the presence of a control group [20,21,26,28,40].

## 4. Discussion

This exhaustive systematic review found conflicting results on the effect of teleworking on MSD, although an increase in nonspecific MSD related to organizational and ergonomic factors seemed to emerge.

Globally, there may be an important effect of the context of teleworking. Several reports have highlighted the strong points and limits of new form of work like teleworking [41,42]. Having a good ergonomic environment, having a good relation between workers and managers, being autonomous at work, and a reasonable workload are all potential critical ergonomic and organizational factors that could prevent the potential negative impact of telework beyond MSD. Several studies in this review reported poor ergonomic work conditions linked to the abrupt change caused by lockdowns [38,39]. A study which focused on workers with a history of LBP found an increase of pain in case of teleworking during lockdown, although this association was not significant when considering workers with a dedicated workstation for teleworking [30]. Another one, though of low evidence, suggested that an ergonomic intervention could lower the risk of MSD [19]. Thus, potential future studies on ergonomic interventions, including information and training, as well as adapted equipment, would be important [43].

Telework brings many advantages, such as the flexibility of workhours or the possibility of work accommodations for people with work limitations. This emphasizes the importance of having data concerning effects on health. The effects of teleworking on other factors, like personal life and work life balance, are also complex, as some studies report positive effects on this point while others report a negative effect with increased technostress and a more blurred border [44,45]. Being sedentary is also an important aspect of teleworking that could influence the risk of MSD, as teleworking decreases break time and activity interruptions, as well as small movements happening during work [45].

A previous integrative review by dos Santos et al. found that musculoskeletal pain increased during the lockdowns, especially in the lower back and neck regions, which could be explained by an increased sedentary lifestyle, poor posture, and increased physical load due to household chores [14]. However, compared to this review, more studies were found due to the updated search, and the risk of bias was assessed. Another review by Oakman et al. on the mental and physical health effects of working at home identified only three studies related to physical health-related outcomes with conflicting conclusions [13].

The heterogenous results of our review could be explained by several factors. First, the studies included were heterogenous, with a varying number of subjects and methods for selecting the participants and populations included. Indeed, some studies recruited participants from universities or public administration [28,32,34,35,36,38], while others recruited participants from private employers [17,18,26,29,33] or from the general population [12,20,21,22,24,25,27,37,39,40]. Second, most studies likely had a high risk of bias, particularly when considering confounding, control groups, and the representativeness of the population studied. Only eight studies considered confounders in the statistical analyses, and among them, six were extensively adjusted for potential confounders [12,20,21,22,26,28]. Most studies also adopted a cross-sectional design, which can lead to reverse causation bias or differential reporting of outcome. Finally, the effect of teleworking of MSD may be different in relation to socioeconomic status and to how telework was implemented. Before the pandemic, workers doing telework were mostly highly qualified and voluntary workers, though from 2022, teleworkers can also be voluntary but also less qualified, with less autonomy. Another result of this systematic review is that all MSD were subjective, for example, the reporting of pain intensity or frequency or reporting change in pain before and after lockdown. There were no medical diagnoses of MSD, like carpal tunnel syndrome or sciatica. However, studies suggest that in the tertiary sector MSD are unspecific, with symptoms like chronic back pain or neck pain [41,42]. The COVID-19 pandemic context is also a potential strong confounder, and most of the studies included assessed the risk of telework on MSD during the pandemic. This systematic review shows the scarcity of evidence of this before the COVID-19 pandemic and shows the need to continue monitoring the effect of telework.

The main limitation of this review is the lack of quantitative analysis and grading. No pulled risk effects and grading were calculated because of the heterogenous methodologies and bias of the included studies. However, our systematic review adopted an exhaustive research protocol on several databases and all articles were selected by at least two different reviewers. The assessment of bias was done in a similar manner, and a category specific to the aim of this review was added. Thus, the qualitative synthetic approach allowed us to highlight the strengths and weaknesses of the different studies assessing the effect of teleworking on MSD. There was a change in the initial protocol for this systematic review in which we aimed to assess the effects of WHF on several health outcomes. However, we quickly focused on MSD, as the number of studies have drastically increased in the last two years. Another potential limit is the lack of MeSH keywords for “Teleworking” before 2021. This could potentially lead to missing studies before this date, but an exhaustive variety of terms were used to retrieve all relevant studies. Lastly, only studies in English or French were considered.

## 5. Conclusions

This systematic review brought to light the necessity of further research to understand the potential effect of teleworking on the risk of MSD. The conceptual model of Beckel and Fisher demonstrates the need for global integrative approaches of teleworking situations that consider the whole work environment in addition to the usual confounders of MSD [46]. MSD and teleworking are challenges that will need to be addressed by researchers and decision makers. Indeed, the flexibility of teleworking and hybrid work seems to be a key factor to promote sustainable work and return to work for workers with disabilities. Future studies should focus on longitudinal approaches and consider ergonomic and work organization factors as well as socio-economic status.

## Figures and Tables

**Figure 1 ijerph-20-04973-f001:**
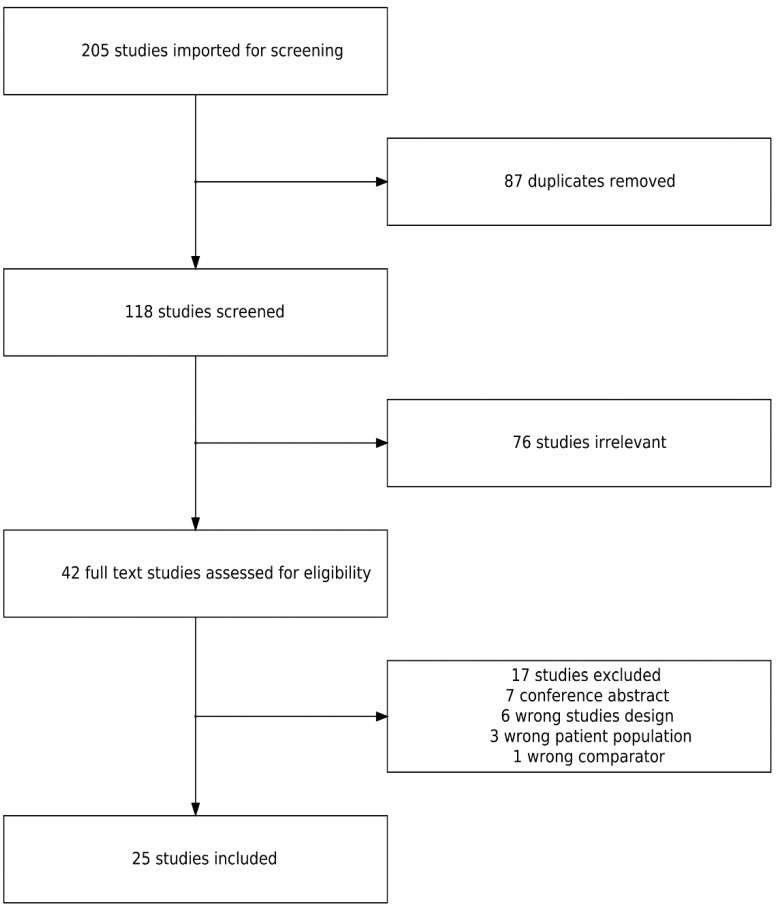
Flow chart of the selection process.

**Table 1 ijerph-20-04973-t001:** Summary of data extracted from selected studies.

Author	Country	Population	Number Included	Definition of MSD Outcome	Confounding	Results	Limitations of Authors
Aegerter, 2021 *	Switzerland	Swiss office workers aged 18–65 years, working more than 25 h per week in sedentary office work, able to communicate in German, in the control cohort between January and April 2020, who answered the COVID-19-related questions in full and were working from home at the time of follow-up	58	NP severity and disability in the last 4 weeks: numeric rating scale and ND index	Fixed effects for workstation ergonomics, working hours at the computer, number of breaks during work, and time	No evidence that ND, number of work breaks, number of hours of computer work changed between pre-COVID-19 (WFH). Evidence of a 0.68-point reduction in NP intensity during the lockdown (95% CI 1.35 to 0.00). Possible increase of the effect of number of hours working on a computer and quality of workplace ergonomics on NP intensity. Possible decrease of the effect of the number of daily work breaks on ND. Strong evidence of poorer workstation ergonomics at home compared to the office	Population in the public sector, social desirability bias, low follow-up, retrospective report of workstation ergonomics at the office at follow-up, no objective criteria for assessing workstation ergonomics
Argus, 2021 *	Estonia	Job described as office work, working with a computer at least 6 h per day, age 18–60 years.	161	Modified NORDIC MSD questionnaire (personalized scale assessing the onset and evolution of the pain)	None	There were no statistically significant differences in the prevalence of MSP before and during the COVID-19 lockdown in different body areas and in total.	Questionnaire-based design, retrospective questions, and absence of data about psychosocial factors and pain intensity
Bailly, 2022 *	France, Switzerland	From the multicenter CONFI-LOMB study regrouping patients from hospitals and one private rheumatology: Adult who undergone a consultation for a common chronic LBP between 1 January 2020, and 17 March 2020 (start of the French lockdown)	360	Change in LBP intensity prior to the lockdown and during lockdown assessed by a 7-point Likert scale	None	In bivariate analyses, LBP increased in the case of teleworking during lockdown (*p* = 0.069) but not in the case of workstation dedicated to teleworking (*p* = 0.249) or equipment adapted to telework (*p* = 0.355)	Cross-sectional design, self-administered and anonymous computerized questionnaire patients from tertiary centers
Deshmukh, 2020 *	India	IT professionals working 100% from home for at least a 6 month period	100	Nordic MSD Questionnaire	None	Results are difficult to interpret because of lack of details	Self-reported data, possibly underestimated their WMSDs to avoid being viewed negatively and small sample size
El Kadri Filho, 2022 *	Brazil	Employees of a Regional Labor Court who were teleworking specifically because of the need for social isolation	55	Nordic MSD questionnaire	None	Regions with the most complaints in the last 6 months and last 7 days were shoulders, neck, and wrists/hands. Posture and job demand exposure assessment was significantly corelated with MSK problems	Only workers in the labor judiciary, data collection eight months after the onset of the pandemic, cross-sectional design
Gerding, 2021 *	USA	All faculty staff, and administration employed by the University of Cincinnati	843	Level of discomfort numeric rating scale for several body regions	None	>40% employees reported moderate to severe discomfort levels in the eyes/neck/head, upper back/shoulders, and lower back regions. Prior to COVID-19, 78.5% experienced little to no discomfort while working in their office setting, and 21.5% had moderate to severe discomfort. Increased glare and lack of having contact with the back of the chair increased discomfort for various body areas	Only staff from one public university, self-reported data on postures and discomfort, retrospective data, and low response rate (10%)
Guler, 2021 *	Turkey?	Workers from companies that started working from home after the pandemic and who practice desk work using a computer	194	Nordic MSD Questionnaire and Visual Analog Scale for work-related pain	Multivariate analysis included independent variables with *p*-values < 0.05 in the univariate analyses	Mean LBP significantly increased during pandemic (WFH) compared to before: 3.14 to 3.56 (*p* = 0.03). LBP was associated with lumbar support before the pandemic, stress level during WFH, general health status, sleep duration, and rest quality during WFH in the multivariate regression model	Cross-sectional design, low number of participants, some specific activities (treadmill, exercise bikes, pilates) were not considered separately, no evaluation of COVID infection during WFH, and no assessment of income changes during WFH
Houle, 2021 *	Canada	Participants aged between 18 and 65 years old and in a full-time telecommuting situation at least one week prior to enrollment	162	NP occurrence, intensity (numerical rating scale) and NP assessed by the Neck Bournemouth Questionnaire (NBQ)	Multivariate analysis with a stepwise method	70% reported at least one NP episode during follow-up. No work-related variables were associated with occurrence of NP episode, including presence of home workstation, headset wearing, telecommuting hours and headset wearing hours. Among telecommuters, NP-related disability was associated with future NP occurrence	Significant attrition following the initial assessment, impossibility to assess other health complaints, no distinction between headache and NP types, small sample size and no assessment of potential factors
Jain, 2022 *	India	University computer users at two Indian universities, over 18 years old, attending regular online classes, performing at least 150–200 min of PA per week	40	Corlett and Bishop’s body part discomfort scale	None	The region with the highest discomfort on the Corlett and Bishop’s scale before intervention was wrist/forearm (8.17 +/− 1.45), lower back (8.01 +/− 1.42), and neck (7.40 +/− 2.71)	Only university students
Knardahl, 2022	Norway	All office workers from private and public organizations recruited from a previous project between 2004–2019	Cross-sectional sample: 7861 Prospective sample: 5258	Reported NP in the last four weeks with a four-level intensity scale and duration of complaint	Adjusted for working more than regular hours, gender, age, skill level, management responsibility, year of measurement (mixed effects regression)	NP not statistically associated with time working at home (hours): aOR and 95% CI compared to 0 h: 0 to 2 h, 0.93 (0.77–1.12), 2 to 5 h, 0.89 (0.71–1.11), 5 to 15, 0.82 (0.63–1.07), more than 15 h, 0.65 (0.38–1.11). Availability expectations was associated with NP	Changes in contents of office jobs between 2004–2020, subjective reports, and cross-sectional design
MacLean, 2022 *	Canada	Staff, administration, and faculty employed by Dalhousie University and primarily working from home with limited on campus access to offices and laboratories since lockdown	445	Changes in work-related discomfort since WFH (Likert scale) and current discomfort and pain using the Nordic MSD Questionnaire and a numeric rating scale	None	61% reported an increase in MSK pain. Area with most reported at least moderate pain were neck, shoulders, and lower back. Seat height and monitor distance were associated with MSK discomfort or pain, respectively β and 95% CI: 6.0 (3.1, 9.0) and 3.3 (0.5,5.9)	Population may be biased (more administrative, women), selection bias, model could not account for certain aspects of home workstation, self-assessed, anonymous WFH ergonomics and perceived musculoskeletal discomfort and pain
Matsugaki, 2021 *	Japan	Population of workers currently in possession of an employment contract who responded that they mainly performed desk work and telecommuted at least once a week	3663	Reported occurrence of stiff shoulder, of LBP in the past 2 weeks, and the average severity of LBP on a numeric rating scale	Adjusted for age, sex, BMI, lifestyle habit (smoking, drinking, PA), number of days with poor mental health (past 30 days), income, educational background, working time, frequency of telecommuting, company size	Telecommuting environment factors (yes/no) associated with LBP were: aOR 95% CI, having enough light on desk 1.43 (1.18–1.73), having enough space to stretch legs 1.30 (1.10–1.54), having a place to concentrate on work 1.38 (1.17–1.64), having appropriate temperature and humidity comfort 1.32 (1.13–1.56), and having enough space on the desk 1.19 (1.02–1.39). Using an office desk or chair was no associated with LBP	History and medication of LBP are unknown, subjective report of home office, lifestyle factors, and working condition, and possibility of a selection bias (more WFH for LBP workers?)
Matsugaki, 2022 *	Japan	Population of workers currently in possession of an employment contract	12,774	Reported occurrence of stiff shoulder, of LBP in the past 2 weeks and the average severity of LBP on a numeric rating scale	Adjusted for age, gender, BMI, marital status, education, income, lifestyle habit (smoking, alcohol, PA), psychological status, company size	LBP was associated with frequency of teleworking aOR 95% CI compared to reference (Almost never): ≤1 d/w 1.18 (0.99–1.41), 2 to 3 d/w 1.27 (1.08–1.50), ≥4 d/w 1.15 (1.01–1.32), *p*-value of trend = 0.003. In a good telecommuting environment, OR did not increase with telecommuting frequency contrary to poor teleworking environment	Subjective assessment of work environment, potential unmeasured confounds, no consideration of duration of workers’ telecommuting engagement and cross-sectional design
Minoura, 2021 *	Japan	Respondents selected from panelists registered within a Japanese Internet survey agency	4227	Occurrence of LBP in the last month and time of appearance (before/after pandemic)	Adjusted for age, sex, weight, education, marital status, having children, outdoor PA during the COVID-19, psychological distress, smoking, alcohol, comorbidities, employment, income level, working time	LBP was associated with increased WFH (yes vs. no) among desk workers during the COVID-19 pandemic: aOR 2.13 (1.52–2.97)	Cross-sectional design, low response rate, potential selection bias, no information on medical LBP diagnosis and quality of work environment, residual confounders, possible time lag between survey and LBP
Moretti, 2020 *	Italy	Population of mobile workers employed as administrative officers that moved to work remotely since the beginning of COVID-19 health emergency	51	Brief Pain Inventory for assessing LBP and NP	None	Main regions for MSK pain were the lower back (41.2%) and the neck (23.5%). Since WFH, NP worsened for 50% participants (N = 6), and 38.1% for LBP (N = 8)	Small sample size, population of a single Italian region, cross-sectional design, some confounders could not be assessed
Muniandy, 2022 *	Malaysia	Lecturers and students of UMS who were actively involved in teaching and learning during the pandemic period	842	Back pain intensity using a numeric rating scale	None	Among newly diagnosed back pain lumbar region was the most frequent (62.1%), and LBP increased after lockdown. Poor ergonomic sitting was associated with mild LBP: OR CI 95% 2.0 (1.2–3.6)	Population of predominantly people higher formal education, survey online accessibility
Oakman, 2022 *	Australia	Participants from across Australia, aged 18 years or older, working from home at least 2 days per week during the period following declaration of the COVID-19 pandemic in Australia, currently living in Australia	924	MSD pain frequency using a 5-point Likert scale and intensity using a 3-point Likert scale	None	After the pandemic, WFH increased for most participants (92.9%). Over 70% reported pain or discomfort at the end of their working day, with a higher level of neck/shoulder pain and hips/legs/feet pain for females	Potential selection bias due to the geographical and gender sample repartition, cross-sectional design
Prieto-González, 2021 *	Slovakia	Pedagogues in Slovakia during the introduction of online classes (January 2021 during COVID-19 pandemic), working in Slovakia in primary, secondary, tertiary, or special needs schools and aged between 18 and 65 years	782	Pain intensity using a numeric rating scale	None	74.84% reported cervical pain and 67.68% reported LBP. The number of days of online classes/week was associated with increased pain intensity: 1/w 3.33 (1.17), 2/w 3.17 (1.07), 3/w 3.46 (1.18), 4/w 3.51 (1.08), and 5/w 3.58 (1.01). Teachers not complying with ergonomic recommendations and sitting most of the time had a higher level of pain intensity	Potential selection bias, detailed medical information available
Radulović, 2021 *	Croatia	Telecommunications company workers working from home for eight months (from 16 March to 4 December 2020) before joining the study	232	Changes in MSD pain before and after WFH using a 3-point Likert scale	None	Among reported LBP, 39.1% had stronger pain when working at home than in the office. Complaint of more severe pain at home than in the office was correlated with not having an ergonomic chair or office desk, longer working hours at home, disturbance at home and women	No limitations reported
Regmi, 2022 *	India	Working population and students across India	1302	Frequency of work-related MSD symptoms while using a digital 4-point Likert scale	None	88% reported work related MSK disorders among which 45% had these symptoms for the first time. MSK symptoms were not statistically associated with WFH (OR 0.61, 95% CI 0.34–1.09), but there were associated with hours of work at home >8 h/d (3.06, 1.89–4.96)	Convenience sampling, students and health care sector population, general health issues and other ocular problems not considered
Rodriguez-Nogueira, 2021 *	Spain	Workers at two Spanish universities employed for at least six months, actively teleworking during the confinement period (between March and May 2020)	472	Nordic MSD Questionnaire adapted to Spanish	None	There was a decrease of pain reported overall during confinement compared to the previous 12 months (*p* < 0.001)	Self-reported stress, pain and PA, cross-sectional design, population of two Spanish universities
Šagát, 2020 *	Saudi Arabia	Resident in Saudi Arabia staying in Riyadh before and during the quarantine, aged between 18 and 64 years	463	Location and intensity of pain on a numerical scale	None	Among subject who WFH or did distance learning during quarantine, there was more MSK pain reported compared to before (48.3% vs. 3.9%). Pain intensity was higher during quarantine than before for subjects WFH or distance learning: mean pain (2.64 vs. 1.97). Subjects who telework or had distance learning had statistically higher MSK pain compared to those who did not (2.63 vs. 2.27)	Certain measurements not included (inflammatory biomarkers, vitamin D levels), assessment of LBP intensity four weeks after confinement, no inclusion of chronic conditions hospitalized patients
Siqueira, 2020 *	Brazil	Brazilian individuals aged between 18 and 59 years	424	Frequency (4-point Likert scale) and MSK pain assessed by the “MSD Pain Investigation Questionnaire”	None	Individuals WFH reported statistically higher frequency of pain than those working in the usual workplace. Individuals WFH had increased frequency of pain during pandemic compared to before in the neck (1.23 vs. 1.05), shoulders (1.33 vs. 1.13), and upper back (1.41 vs. 1.20)	Potential reporting bias on data before pandemic
Tezuka, 2022 *	Japan	Full-time workers of the two non-ferrous metal companies, aged 20 years or more, and non-teleworker before the emergency declaration	917	Presence of physical symptoms (from leading symptoms in the National Lifestyle Survey of Japan) not due to COVID-19 infection during the emergency	Adjusted for age, sex, BMI, marital status, occupational status, and stiff shoulders before the emergency declaration	Telework frequency was statistically associated with LBP during emergency declaration: compared to 0 days of telework, aOR and 95% CI, 1–2/w 3.83 (1.41–10.36), 3–4/w 6.09 (2.33–15.94), 5 or more/w 5.57 (2.22–14.00)	Cross-sectional design, potential recall bias, potential selection bias due to the low response rate (34.6%), high percentage of men in the sample and survey distribution method
Widianawati, 2020 *	Indonesia	WFH workers during the COVID-19 pandemic in Indonesia	50	MSD pain assessed by a numerical scale	None	28% of workers reported complaints of low MSD with an average pain value of 50.44. The most frequent regions for pain were the neck and the lower back	No limitations reported

BMI: body mass index, ND: neck disability, NP: neck pain, LBP: LBP: low back pain, WFH: work from home, OR: odds ratio, MSD: musculoskeletal disease, PA: physical activity. * During COVID-19 pandemic studies.

**Table 2 ijerph-20-04973-t002:** Assessment of risk of bias according to the Navigation Guide for systematic reviews in environmental and occupational health [16].

Name	1. Are the Study Groups at Risk of Not Representing Their Source Populations in a Manner That Might Introduce Selection Bias?	2. Was Knowledge of the Group Assignments Inadequately Prevented (i.e., Blinded or Masked) during the Study, Potentially Leading to Subjective Measurement of Either Exposure or Outcome?	3. Were Exposure Assessment Methods Lacking Accuracy?	4. Were Outcome Assessment Methods Lacking Accuracy?	5. Was Potential Confounding Inadequately Incorporated?	6. Were Incomplete Outcome Data Inadequately Addressed?	7. Does the Study Report Appear to Have Selective Outcome Reporting?	8. Did the Study Receive any Support from a Company, Study Author, or Other Entity Having a Financial Interest in any of the Exposures Studied?	9. Did the Study Appear to Have Other Problems That Could Put It at a Risk of Bias? Population Studied Does Not Represent General Teleworker Population or No Control Group
Aegerter 2021	Probably low	Not applicable	Probably low	Low	Probably low	Probably low	Not applicable	Low	Probably low
Argus 2021	Probably low	Not applicable	Probably low	Low	Probably high	Probably high	Not applicable	Low	Probably high
Bailly 2022	Low	Not applicable	Probably low	Probably low	Probably high	Probably low	Not applicable	Low	Probably high
Deshmukh 2020	High	Not applicable	Probably low	Low	High	Probably low	Not applicable	Probably low	High
El Kadri Filho 2022	Low	Not applicable	Low	Low	High	Probably low	Not applicable	Low	Probably high
Gerding 2021	High	Not applicable	Probably high	Probably high	Probably high	Probably low	Not applicable	Low	Probably high
Guler 2021	Probably low	Not applicable	Low	Low	Probably high	Probably high	Not applicable	Low	Probably high
Houle 2021	Probably high	Not applicable	Probably low	Low	Probably high	Probably high	Not applicable	Low	High
Jain 2022	Probably high	Not applicable	Probably high	Low	Probably high	Probably high	Not applicable	Low	High
Knardahl 2022	Low	Not applicable	Probably low	Probably low	Low	Probably low	Not applicable	Low	Probably high
MacLean 2022	Probably high	Not applicable	Probably low	Low	Probably low	Probably low	Not applicable	Low	Probably high
Matsugaki 2021	Low	Not applicable	Low	Low	Low	Probably low	Not applicable	Low	Probably low
Matsugaki 2022	Low	Not applicable	Low	Probably low	Low	Probably low	Not applicable	Low	Low
Minoura 2021	Low	Not applicable	Probably low	Probably low	Low	Probably low	Not applicable	Low	Probably high
Moretti 2020	Probably high	Not applicable	Probably low	Low	High	Probably low	Not applicable	Low	Probably high
Muniandy 2022	Probably high	Not applicable	Probably high	Probably low	Probably high	Probably low	Not applicable	Low	Probably high
Oakman 2022	Probably high	Not applicable	Probably low	Probably low	High	Probably high	Not applicable	Low	Probably high
Prieto-Gonzalez 2021	Probably high	Not applicable	Probably high	Probably high	Probably high	Low	Not applicable	Low	Probably high
Radulovic 2021	Probably low	Not applicable	Probably high	Probably high	High	Low	Not applicable	Probably low	Probably high
Regmi 2022	Probably high	Not applicable	Probably low	Probably low	Probably high	Probably low	Not applicable	Low	High
Rodriguez-Nogueira 2021	Probably high	Not applicable	Probably low	Low	Probably high	Probably low	Not applicable	Low	Probably high
Sagat 2020	Low	Not applicable	Probably high	Probably low	Probably high	Probably low	Not applicable	Low	Probably high
Siqueira 2020	Probably high	Not applicable	Probably low	Probably high	High	Probably low	Not applicable	Low	Probably low
Tezuka 2022	Probably high	Not applicable	Low	Probably high	Low	Probably low	Not applicable	Probably low	Probably low
Widianawati 2020	Probably high	Not applicable	Probably low	Probably high	High	Low	Not applicable	Probably low	High

Risk of bias assessment,: red cell = high, orange cell = probably high, light green = probably low, dark green = low.

## Data Availability

The data presented in this study are available in the article.

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
