# Peer review of "Teleworking and Musculoskeletal Disorders: A Systematic Review"

_ijerph, 2023, doi:10.3390/ijerph20064973_

Round 1

Reviewer 1 Report

Major concerns:

1. Please, mention the general aspects of teleworking pointing out both, positive and negative effects of this type of work.

2. The authors should pay attention to a problem of scarce evidence about the MSD in teleworkers before the COVID pandemic.

3. Please, discuss the problem of lack of ergonomic adjustment of the workers’ equipment in their homes during COVID and teleworking. Generally, during COVID, the workers were moved to teleworking without any training about ergonomic aspects of teleworking. Additionally, most of the teleworkers have worked on their own computers and other electronic devices (phones, etc.), in their non-adopted home conditions, even if the workplace in the office was fully adapted to the teleworking.

4. Please, discuss the problem of aggravation of MSD during lockdowns and COVID pandemic, especially, in handicapped persons, who were teleworkers before COVID period.

5. Is it possible to assess the frequency of pain related disorders in teleworkers? (was it everyday illness?, every second day?).

6. Please, discuss some possible ways of spending the spare time in teleworkers during COVID-related lockdowns. It is essential, because the sedentary lifestyle during lockdowns and quarantines potentiated the MSD in teleworkers and their family members. 

Minor concerns:

1. Table 1 should be moved to the Supplementary Materials as a Suppl. Table 1.

2. The third column of the Table 1 “During COVID”. Please, remove this column and insert the necessary information to the legend to the table, in form of an asterisk. Almost all publications (except for 1 on page 8) were “during COVID”.

3. Line 132 – it should be cited Table 2.

4. Line 129 - explain the "MSK" acronym.

Author Response

Major concerns:

  1. Please, mention the general aspects of teleworking pointing out both, positive and negative effects of this type of work.

R1.1. We have developed the potential negative and positive effect of telework and added relevant reference in the introduction and discussion.

“The effect of teleworking on health seems to be contrasted, both positive and negative depending on the situations, with a predominant role of contextual factors. Even before the pandemic happened, teleworking was implemented as a useful tool for allowing sustainable work and return to work especially for workers with disabilities or suffering from chronic diseases, including cancer. Indeed, traditional work environments may present barriers for these workers, especially those with cognitive limitations, and an adapted and familiar work environment at home may facilitate employment. Such accommodations require many changes in the work culture and vary significantly between countries. Bouziri et al. highlighted the potential health impact during the COVID-19 pandemic as well as possible recommendations for de-creasing the new related risk. For example, during containment, telework decreased the risk associated with transportation but also showed the lack of ergonomic measures for home workstations and the work environment in general.”

  1. The authors should pay attention to a problem of scarce evidence about the MSD in teleworkers before the COVID pandemic.

R1.2. We definitely agree with the reviewer as this systematic review points out that very few studies assessed the risk of MSD before the COVID-pandemic. We have added this point in the discussion.

“The COVID-19 pandemic context is also a potential strong confounder and most of the studies included assessed the risk of telework on MSD during the pandemic. This systematic review shows the scarcity of evidence of this of relation before the COVID-19 pandemic and the need to continue monitoring the effect of telework.”

  1. Please, discuss the problem of lack of ergonomic adjustment of the workers’ equipment in their homes during COVID and teleworking. Generally, during COVID, the workers were moved to teleworking without any training about ergonomic aspects of teleworking. Additionally, most of the teleworkers have worked on their own computers and other electronic devices (phones, etc.), in their non-adopted home conditions, even if the workplace in the office was fully adapted to the teleworking.

R1.3. At the reviewer’s advice, we have highlighted this point in the discussion. Indeed, some of the included studies have found poor ergonomic adjustment at home and some even suggested that good ergonomic could decrease the association between MSD and telework.

“Several studies in this review reported poor ergonomic work conditions linked to the abrupt change caused by lockdowns. One study, though of low evidence, suggested that an ergonomic intervention could lower the risk of MSD. Thus, potential future studies on ergonomic interventions including information and training as well as adapted equipment would be important.”

  1. Please, discuss the problem of aggravation of MSD during lockdowns and COVID pandemic, especially, in handicapped persons, who were teleworkers before COVID period.

R1.4. There were few studies that focused on previously handicapped persons. We have detailed this point in the discussion.

 “A study which focused on workers with history of LBP found an increase of pain in case of teleworking during lockdown, though this association was not significant when considering workers with a dedicated workstation for teleworking.”

  1. Is it possible to assess the frequency of pain related disorders in teleworkers? (was it everyday illness? every second day?).

R1.5. Unfortunately, probably in relation to the abrupt changes caused by the pandemic, definition of the outcomes was various (see Table 1). Most of them inquired whether there was pain in the last 7 days or 1 month or 6 months etc.

  1. Please, discuss some possible ways of spending the spare time in teleworkers during COVID-related lockdowns. It is essential, because the sedentary lifestyle during lockdowns and quarantines potentiated the MSD in teleworkers and their family members.

R1.6. Indeed, teleworking has many effects on other factors including sedentarity and the boundary between work and personal life. We have developed this point in the discussion, while keeping in mind that there were not the main focus of this review.

“The effects of teleworking on other factors, like the personal life and work life balance are also complex as some studies report positive effects on this point while others report negative effect with increased technostress and a more blurred border. Sedentarity is also an important aspect of teleworking that could influence the risk of MSD as tele-working decreases break time and activity interruptions as well as small movements happening during work”.

Minor concerns:

  1. Table 1 should be moved to the Supplementary Materials as a Suppl. Table 1.

R1.7. We understand that Table 1 was too big for the main manuscript. However, we believe that added a short summary of the evidence is relevant, so we adapted a “short” version of Table 1 in the manuscript and moved the “full” version to supplementary material 2.

“An exhaustive summary of the studies is available in Supplementary material 2.”

  1. The third column of the Table 1 “During COVID”. Please, remove this column and insert the necessary information to the legend to the table, in form of an asterisk. Almost all publications (except for 1 on page 8) were “during COVID”.

R1.8. We have removed this column and added a footnote corresponding to studies underwent during the COVID-19 pandemic.

  1. Line 132 – it should be cited Table 2.

R1.9. We have cited the table as “Table 2”.

“Table 2 shows the risk of bias assessment for all the studies included.”

  1. Line 129 - explain the "MSK" acronym.

R1.10. We thank the reviewer for pointing out the mistake. We have fully worded the acronym MSK as “musculoskeletal”.

Reviewer 2 Report

The reviewed article aimed to make a qualitative systematic review 12 on the effect of teleworking on MSD.

The stated goal has been met.

I recommend enriching the content of the article by expanding the introduction. It is worth defining teleworking and musculoskeletal disorders more broadly.

I suggest expanding the concluding remarks, as they should be more comprehensive.

In addition, I propose checking certain aspects of bibliographic reference lists. For example, the journal’s name should be written with its abbreviation, the volume in italics, and the year in bold, and the text should be justified. There are several shortcomings. It needs improvement.

Author Response

  1. I recommend enriching the content of the article by expanding the introduction. It is worth defining teleworking and musculoskeletal disorders more broadly.

R2.1. We have expanded on the definition of telework and musculoskeletal disorders in the introduction, as well as added relevant references.

“There are several terms related to this type of work including remote work, which is the broadest term where the workplace can be anywhere outside the usual place, telework which implies the usage electronic devises for remote work and home-based work/work from home (WFH) which can imply that the default working place is at home, including work performed at home and hybrid work which is a combination of work from home (WFH) and work in the office. Hybrid work is a growing form of work which com-bines WFH and work in the office.”

“MSD is a broad term that reference at any affection of the soft periarticular tissues, following the EU-OSHA (European Agency for Safety and Health at Work). Medically codified diseases are included in this definition, like carpal tunnel syndrome or rotator cuff disease, but also unspecific conditions like low back pain or neck pain”.

  1. I suggest expanding the concluding remarks, as they should be more comprehensive.

R2.2. We have followed the reviewer’s advice and expanded the concluding remark which now includes notion of integrative approaches to assess health effect linked to telework as well as the potential of sustainable work for workers with disabilities.

“The conceptual model of Beckel and Fisher demonstrate the need of global integrative approaches of teleworking situations that consider the whole work environment in addition to the usual confounders of MSD. […] Indeed, the flexibility of teleworking and hybrid work seems to be a key factor to pro-mote sustainable work and return to work for workers with disabilities.”

  1. In addition, I propose checking certain aspects of bibliographic reference lists. For example, the journal’s name should be written with its abbreviation, the volume in italics, and the year in bold, and the text should be justified. There are several shortcomings. It needs improvement.

R2.3. We thank the reviewer for point out this shortcoming. We have corrected the references format according to the journal’s guidelines.

Reviewer 3 Report

The number of references in the introduction is scarce, perhaps some more should be included, either related to the lockdown or to teleworking.

Author Response

  1. The number of references in the introduction is scarce, perhaps some more should be included, either related to the lockdown or to teleworking.

R3.1. We have expanded the introduction and the discussion and added relevant references related to teleworking and lockdowns (see marked revised manuscript).

Round 2

Reviewer 1 Report

No further comments on the article